# KNOWLEDGE BASED MULTILINGUAL LANGUAGE MODEL

## ABSTRACT

Knowledge enriched language representation learning has shown promising performance across various knowledge-intensive NLP tasks. However, existing knowledge based language models are all trained with monolingual knowledge graph data, which limits their application to more languages. In this work, we present a novel framework to pretrain knowledge based multilingual language models (KMLMs). We first generate a large amount of code-switched synthetic sentences and reasoning-based multilingual training data using the Wikidata knowledge graphs. Then based on the intra- and inter-sentence structures of the generated data, we design pretraining tasks to facilitate knowledge learning, which allows the language models to not only memorize the factual knowledge but also learn useful logical patterns. Our pretrained KMLMs demonstrate significant performance improvements on a wide range of knowledge-intensive cross-lingual NLP tasks, including named entity recognition, factual knowledge retrieval, relation classification, and a new task designed by us, namely, logic reasoning. Our code and pretrained language models will be made publicly available.

## 1 INTRODUCTION

Pretrained language models (PTLMs) such as BERT (Devlin et al., 2019) and RoBERTa (Liu et al., 2019) have achieved superior performances on a wide range of natural language processing (NLP) tasks. Existing PTLMs usually learn universal language representations from general-purpose large-scale corpora but lack the ability to capture world's factual knowledge. It has been shown that structural knowledge data can provide rich factual information for better language understanding. A number of studies have demonstrated the effectiveness of incorporating such factual knowledge from external knowledge bases into PTLMs (Zhang et al., 2019; Liu et al., 2020; Peters et al., 2019; Poerner et al., 2020; Wang et al., 2021a). However, existing knowledge-enriched LMs are all trained with monolingual knowledge graph data, which limits their applications to other languages.

In this work, we attempt to explore knowledge-enriched pretraining under a multilingual setting, which is rarely explored in prior work. There are several motivations. First, a multilingual version of knowledge-enriched PTLM allows a larger range of population to benefit from the knowledge-enriched representations. Second, we argue that existing multilingual pretraining framework can benefit from incorporating structured multilingual knowledge. Knowledge is language-agnostic, i.e. different languages share almost the same entities and relations. We can opportunistically obtain a huge amount of aligned data across different languages from existing knowledge graphs. Incorporating this information into the multilingual PTLMs not only helps the model better capture the factual knowledge but also improves its cross-lingual capability.

We propose KMLM, a novel framework for **K**nowledge-based **M**ultilingual **L**anguage **M**odel Pretraining. Unlike prior knowledge-injection approaches (Zhang et al., 2019; Liu et al., 2020; Peters et al., 2019; Wang et al., 2021a), the proposed framework requires neither structured knowledge encoding to encode entities or relations with a separate encoder, nor heterogeneous information fusion to fuse multiple types of embeddings (e.g. entities and words). The key idea of KMLM is to convert the structured knowledge from knowledge graphs to sequential data which can be directly fed as input to the language model pretraining. Specifically, we generate two types of training data – the *code switched* knowledge data and the *reasoning based* data. The former is obtained by generating code switched sentences from triplets of the multilingual knowledge graph Wikidata (Vrandečić &

Krötzsch, 2014). The later is constructed by converting cycles from Wikidata knowledge graph into word sequences in different languages, which contain rich logical patterns. We then propose two pretraining tasks that are operated on the code switched and reasoning based data, respectively. One is designed to memorize the factual knowledge across languages, and the other is to learn the logical patterns by reasoning.

Compared to existing knowledge-enriched pretraining methods (Zhang et al., 2019; Liu et al., 2020; Peters et al., 2019), KMLM has the following key advantages. (1) It is explicitly trained to derive new knowledge through logical reasoning. Therefore, in addition to memorizing knowledge facts, it also learns the logical patterns from the data. (2) KMLM does not require a separate encoder for knowledge graph encoding. Furthermore, it does not rely on any entity linker to link the text to the corresponding entities, as done in existing methods (Zhang et al., 2019; Peters et al., 2019; Poerner et al., 2020). (3) KMLM keeps the model structure of the multilingual PTLM without introducing any additional component during both training and inference stages. This makes the training much easier and the trained model is directly applicable to downstream NLP tasks.

We evaluate KMLM on a wide range of knowledge-intensive cross-lingual NLP tasks, including named entity recognition, factual knowledge retrieval, relation classification, and logic reasoning. The novel logic reasoning task is designed by us to test the reasoning capability of the models. Our KMLM achieves consistent and significant improvements on all knowledge-intensive tasks, meanwhile it does not sacrifice the performance on general NLP tasks.

## 2 RELATED WORK

Knowledge-enhanced language modeling aims to incorporate knowledge, concepts and relations into the pretrained language models (Devlin et al., 2019; Liu et al., 2019; Brown et al., 2020), which proved to be beneficial to language understanding (Talmor et al., 2020a).

The existing approaches can be roughly divided into two lines: implicit knowledge modeling and explicit knowledge injection. Previous attempts on implicit knowledge modeling usually consist of entity-level masked language modeling (Sun et al., 2019; Liu et al., 2020), entity-based replacement prediction (Xiong et al., 2020), knowledge embedding loss as regularization (Wang et al., 2021b) and universal knowledge-text prediction (Sun et al., 2021). In contrast to implicit knowledge modeling, the methods of explicit knowledge injection separately maintain a group of parameters for representing structural knowledge. Such methods usually require a heterogeneous information fusion component to fuse multiple types of embeddings obtained from the text and knowledge graphs.

Zhang et al. (2019); Poerner et al. (2020) employ external entity linker to discover the entities in the text and perform feature interaction between the token embeddings and entity embeddings during the encoding phase of a transformer model. Peters et al. (2019) borrow the pre-computed knowledge embeddings as the supporting features of training an internal entity linker. Wang et al. (2021a) take the advantage of adapter (Houlsby et al., 2019) and insert an adapter component in each transformer layer to store the learned factual knowledge. Despite the effectiveness of these methods on knowledge-intensive tasks, the research for jointly modeling knowledge and pretraining multilingual language model is still left blank. Besides, the above studies only focused on memorizing the existing facts but ignored the reasoning over the unseen/implicit knowledge that is derivable from the existing facts. Such reasoning capability is regarded as a crucial part of building consistent and controllable knowledge-based models (Talmor et al., 2020b).

In this paper, with the purpose of consolidating knowledge modeling and multilingual pretraining (Mulcaire et al., 2019; Conneau et al., 2020), and boosting the capability of implicit knowledge reasoning, we explore the solutions for multilingual knowledge-enhanced pretraining.

## 3 FRAMEWORK

In this section, we describe the proposed framework for knowledge based multilingual language model (KMLM) pretraining. We first generate a large amount of knowledge-intensive multilingual training data from the Wikidata (Vrandečić & Krötzsch, 2014) knowledge base, and then design tasks to train the language models to memorize factual knowledge and learn logical patterns from the generated data.

## 3.1 KNOWLEDGE INTENSIVE TRAINING DATA

In addition to the large-scale plain text corpus that is commonly used for language model pretraining, we also generate a large amount of knowledge intensive training data from Wikidata (Vrandečić & Krötzsch, 2014), a publicly accessible knowledge base by collaborative editing. Wikidata is composed of a large set of knowledge graph triplets $(h, r, t)$, where $h$ and $t$ are the head and tail entities respectively, $r$ is the relation type. As shown by an example in Table 1, most of the entities, as well as the relations in Wikidata, are annotated in multiple languages, and in each language, many aliases are also given, although they are usually less frequently used.

Table 1: An example of the Wikidata entity labels and aliases in multiple languages. This example is taken from `https://www.wikidata.org/wiki/Q1420`. Q1420 is the unique entity ID.

| ID | Language | Label | Aliases |
|---|---|---|---|
| Q1420 | English | motor car | auto, autocar, automobile, motorcar, car, … |
| | Spanish | automóvil | coche, carro, auto, automovil, … |
| | Hungarian | autó | gépkocsi, automobil, személygépkocsi, személyautó, … |
| | … | … | … |

Original (en):
(motor car, designed to carry, passenger)

Code Switched (en-fr):
motor car [mask] **conçu pour transporter** [mask] passenger.
motor car [mask] designed to carry [mask] **passager**.
…

Code Switched (en-fr) & Alias Replaced:
automobile [mask] **destiné au transport** [mask] passenger.
motor car [mask] intended to carry [mask] **passager**.
…

Figure 1: Examples of the en-fr code switched synthetic sentences. The words replaced with translations or aliases are marked with bold font and underline, respectively.

**Code Switched Synthetic Sentences** Training language models on high quality code switched sentences is one of the most intuitive ways to learn language agnostic representations (Winata et al., 2019), where the translations of words/phrases can be treated in a similar way as their aliases. Meanwhile, the code mixing techniques have proved to be helpful for improving cross-lingual transfer performance in many NLP tasks (Qin et al., 2020; Santy et al., 2021). Therefore, we propose a novel method to generate code switched synthetic sentences using the multilingual knowledge graph triplets. See Fig. 1 for some generated examples.

For each triplet $(h, r, t)$ in Wikidata, we use $h_{l,0}$ to denote the default label of $h$ in language $l$. For the entity Q1420 in Table 1, $h_{en,0}$ is "motor car" and $h_{es,0}$ is "automóvi". $h_{l,i}$ denotes the aliases when the integer $i > 0$. We define $r_{l,i}$ and $t_{l,i}$ in the same way for the relation and the tail entity, respectively. Since English is resource-rich and often used as the source language for cross-lingual transfer, we only consider language pairs of $\{(en, l')\}$ for code switching, where $l'$ is any other none English language. With such design, English can also work as a bridge for cross-lingual transfer between a pair of none English languages.

Specifically, the code-switched synthetic sentences for $(h, r, t)$ can be generated in 4 steps: 1) Determining the language pair $(en, l')$; 2) Finding the English default labels $(h_{en,0}, r_{en,0}, t_{en,0})$; 3) For each item in the triplet, uniformly sample a value $v \in \{true, false\}$, if $v = true$ and the item has translation (i.e. default label) in $l'$, then replace the item with the corresponding translation in $l'$; 4) Generate the sequence of *"h [mask] r [mask] t."* by inserting two mask tokens. The alias replaced

sentences can be generated in a similar way, except that we random sample aliases in the desired language to replace the default label in steps 2 and 3.

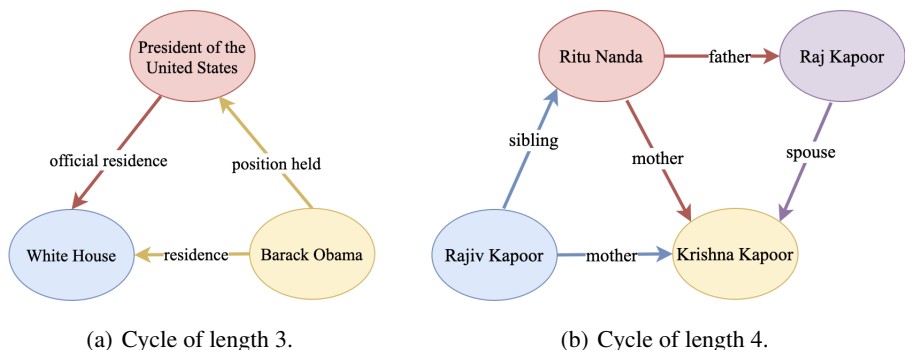

(a) Cycle of length 3.    (b) Cycle of length 4.

Figure 2: Examples of the cycles extracted from knowledge graph.

**Reasoning Based Training Data**   The logical reasoning ability allows humans to solve more complex problems with limited information. However, this ability was not given enough attention when training the previous language models. In knowledge graphs, we can use nodes to represent entities, and use edges between any two nodes to represent their relations. In order to train the model to learn logical patterns, we generate a large amount of reasoning based training data by finding cycles from the Wikidata knowledge graph. As shown with an example in Fig. 2(a), the cycles of length 3 can be viewed as the basic components for more complex logical reasoning process. We train language models to learn the co-occurrence of the relations and entities so as to infer the best candidate relations for incomplete cycles, i.e. deriving the implicit information from the given context.

Similar to the structure of the code switched synthetic sentences discussed above, the cycles in Fig. 2(a) is composed of 3 triplets, and hence can be converted to 3 synthetic sentences (the first example shown in Fig. 4). To increase the difficulty, we also extract cycles of length 4 to generate the reasoning oriented training data. However, we find simply increasing the length of cycles will make the samples less logically coherent, so we require the cycle to have at least one additional diagonal edge. Fig. 2(b) shows such an example. It can be converted to a training sample of 5 sentences in the same way as above. For the multilingual reasoning based data, we only generate monolingual sentences, i.e. without applying code mixing.

We treat Wikidata as an undirected graph when extracting cycles. Given an entity, the length-3 cycles containing this entity can be easily extracted by first finding all of the neighbour entities, and then iterating through the pairs of neighbour entities to check whether they are also connected. The length-4 cycles with an additional diagonal edge connecting the two neighbours can be extracted with a few extra steps. Assuming we have identified a length-3 cycle containing entity $A$ and its two neighbour entities $B$ and $C$, we can iterate through the neighbours of $B$ (excluding $A$ and $C$) to check whether it is also connected to $C$. We also need to remove duplicates when generating a large amount of cycles.

## 3.2   PRETRAINING TASKS

**Multilingual Knowledge Oriented Pretraining**   In the generated synthetic sentences, the *"[mask]"* tokens are added between entities and relations to denote the linking words. For example, the first mask token in *"motor car [mask] designed to carry [mask] **passager**."* may denote *"is"*, while the second one may denote *"certains"*.[1] Since the ground truth of such masked linking words are not known, we do not compute the loss of those corresponding predictions. Instead, we randomly mask the remaining tokens in the code switched synthetic sentence, and compute the cross entropy loss $\mathcal{L}_{CS}$ over these masked entity and relation tokens (Fig. 3).

---

[1]"certains" in French means "some" or "certain".

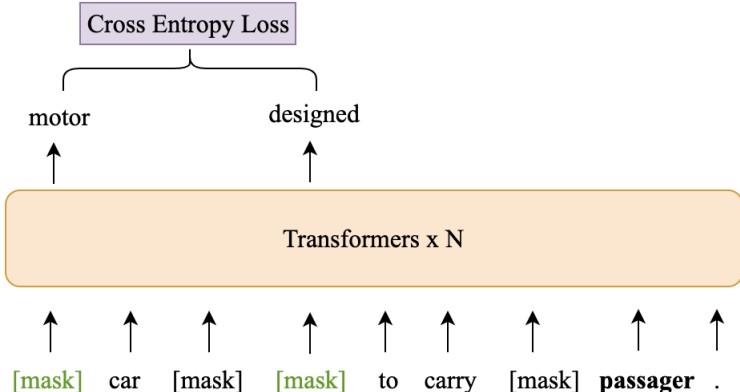

Figure 3: MLM on the code switched synthetic sentence *"motor car [mask] designed to carry [mask] passager."*. The cross entropy loss $\mathcal{L}_{CS}$ is only computed over the randomly masked entity and relation tokens highlighted in lime green. For simplicity, the sub-word tokens are not shown in this example.

**Length-3 Cycle:**
Original: President of the United States [mask] official residence [mask] White House. Barack Obama [mask] residence [mask] White House. Barack Obama [mask] position held [mask] President of the United States.

Masked: President of the United States [mask] official residence [mask] White House. Barack Obama [mask] [mask] [mask] White House. Barack Obama [mask] position held [mask] President of the United States.

**Length-4 Cycle:**
Original: Ritu Nanda [mask] father [mask] Raj Kapoor. Ritu Nanda [mask] mother [mask] Krishna Kapoor. Rajiv Kapoor [mask] mother [mask] Krishna Kapoor. Rajiv Kapoor [mask] sibling [mask] Ritu Nanda. Raj Kapoor [mask] spouse [mask] Krishna Kapoor.

Masked: Ritu Nanda [mask] father [mask] Raj Kapoor. Ritu Nanda [mask] mother [mask] [mask] [mask]. Rajiv Kapoor [mask] mother [mask] Krishna Kapoor. Rajiv Kapoor [mask] [mask] [mask] Ritu Nanda. Raj Kapoor [mask] spouse [mask] Krishna Kapoor.

Figure 4: Examples of the masked training samples for logic reasoning. The relations are highlighted in orange. The masked entity and relation tokens are highlighted in lime green.

**Logical Reasoning Oriented Pretraining** We design tasks to train the model to learn logical reasoning patterns from the synthetic sentences generated from the length-3 and length-4 cycles. As the examples shown in Fig. 4, we convert the relation and entity prediction tasks into the same format as masked language modeling. For the length-3 cycles, each entity appears exactly twice in every training sample. If we design task to predict masked entities, it can be easily solved by counting the appearance numbers of the entities. Therefore, we only mask one random relation in each sample for model training, and let the model learn to predict the masked relation tokens based on the context.

Two types of tasks are designed to train the model to learn reasoning with the length-4 cycles: 1) For 80% of the time, we train the model to predict randomly masked relation and entities. We first mask one random relation. To increase the difficulty, we also mask one or two randomly selected entities at equal chance. The lower half of Fig. 4 shows an example where one relation and one entity are masked. 2) For the remaining 20% of the time, we randomly mask a whole sentence to let the model learn to derive new knowledge from the remaining context. To provide some hints on the expected new knowledge, we keep the relation of the selected sentence unmasked, i.e., only mask its two entities. The logical reasoning task loss $\mathcal{L}_L$ can also be computed with the cross entropy loss over the masked tokens. Note that masked entity prediction is not always non-trivial in this task. For example, when we mask exactly one entity and the entity $E$ only appears once in the masked sample, then it is easy to guess $E$ is the masked one. In Fig. 4, a concrete example is masking the first

appearance of *"Raj Kapoor"* in the original sentence of the length-4 cycle. We do not deliberately avoid such cases, since they may help introduce more diversity to the training data.

**Loss Function** In addition to the pretraining tasks designed above, we also train the model on the plain text data with the original masked language modeling loss $\mathcal{L}_{MLM}$ used in previous works (Devlin et al., 2019; Conneau et al., 2020). Therefore, the final loss can be computed as:

$$\mathcal{L} = \mathcal{L}_{MLM} + \alpha(\mathcal{L}_{CS} + \mathcal{L}_L), \tag{1}$$

where $\alpha$ is the hyper-parameter to adjust the weights of the original MLM and the losses for modeling the multilingual knowledge and logical reasoning.

## 4 EXPERIMENTS

In this section, we first describe the pretraining details of our KMLMs. Then we verify its effectiveness on the knowledge-intensive NLP tasks. Finally, we examine its performance on some general cross-lingual NLP tasks.

### 4.1 PRETRAINING DETAILS

Our proposed framework can be conveniently implemented on top of the existing transformer encoder based language models like mBERT (Devlin et al., 2019) and XLM-R (Conneau et al., 2020) without any modification of the model structure. Therefore, instead of pretraining the model from scratch, it is more efficient and cost saving to initialize the model with the checkpoints of existing pretrained models. In this experiment, we build our knowledge intensive training data in 10 languages, including English (en), Vietnamese (vi), Dutch (nl), German (de), French (fr), Italian (it), Spanish (es), Japanese (ja), Korean (ko) and Chinese (zh). We only use the 5 million entities and 822 relations filtered by Wang et al. (2021b), and generate 250 million code switched synthetic sentences[2] and 100 million reasoning based training samples following the steps described in §3.1. In addition, 260 million sentences are also sampled from the CC100 corpus[3] (Wenzek et al., 2020) for the 10 languages.

Our models KMLM$_R$-XLM-R$_{BASE}$ and KMLM$_R$-XLM-R$_{LARGE}$ are initialized with the XLM-R base and large checkpoints, respectively. Then we continue to pretrain these models with the proposed tasks (§3.2). The subscript $R$ is used to denote the models trained with the logical reasoning tasks. In addition, we also train a variant, KMLM-XLM-R$_{BASE}$, on the same data but without the logical reasoning tasks, that is when training this model, the masked language modeling task is also used on the reasoning based data. Previous studies showed the original mBERT model outperforms XLM-R on the X-FACTR (Jiang et al., 2020) and RELX (Köksal & Özgür, 2020) tasks, so we also train a KMLM$_R$-mBERT model with the mBERT base checkpoint as initialization. Since the original mBERT is trained using the Wikipedia corpus, for a more faithful comparison, we replace the CC100 plain text with sentences sampled from Wikipedia to train KMLM$_R$-mBERT. To speed up pretraining, we use the max sequence length of 128 to train the models first, and then finetune them on plain text with max the sequence length of 512 for another 600 steps. See Appendix § A.1 for the detailed statistics of the training data and hyper-parameters used for pretraining.

### 4.2 CROSS-LINGUAL NAMED ENTITY RECOGNITION

Named entity recognition (NER) involves identifying and classifying named entities from unstructured text data. Our generated training data contains a large amount of entities, and describes their relations more concisely compared with the natural sentences. This may allow the language models to directly memorize these entities during pretraining and learn their similarities and differences more efficiently for better NER performance.

We conduct experiments on Conll02/03 (Tjong Kim Sang, 2002; Tjong Kim Sang & De Meulder, 2003) and WikiAnn (Pan et al., 2017) NER data to verify the effectiveness of our framework. For convenience, the WikiAnn data processed by Hu et al. (2020) is used in our experiment. We use the

---

[2]Include 125 million code switched sentences and 125 million alias replaced sentences.
[3]http://data.statmt.org/cc-100/

same transformer based NER model and hyper-parameters as Hu et al. (2020) to finetune and select the best models with the English train and dev splits, respectively. Finally, the NER models are evaluated on the target language test sets to check their zero-shot cross-lingual NER performance. All of the results are averaged over runs with 3 different random seeds. The baseline model results are reproduced in the same environment.

The results on Conll02/03 data are presented in Table 2. Compared with XLM-R$_{BASE}$, our model KMLM-XLM-R$_{BASE}$ improves the average F1 on all 4 languages by 1.75 points. Especially on German, our model demonstrates 4.89 points improvement. There is no obvious performance difference between KMLM-XLM-R$_{BASE}$ and KMLM$_R$-XLM-R$_{BASE}$ because NER may not need logical reasoning. Even when compared with XLM-R$_{LARGE}$, our large model still improves the average performance by 0.92.

Table 2: Zero-shot cross-lingual NER F1 on the Conll02/03 datasets.

|  | en | de | nl | es | avg. |
|---|---|---|---|---|---|
| XLM-R$_{BASE}$ | 91.16 | 68.87 | 79.00 | **76.70** | 78.93 |
| KMLM-XLM-R$_{BASE}$ (ours) | 91.38 | **73.76** | **81.55** | 76.03 | **80.68** |
| KMLM$_R$-XLM-R$_{BASE}$ (ours) | **91.47** | 73.52 | 80.95 | 76.59 | 80.63 |
| XLM-R$_{LARGE}$ | **92.98** | 73.79 | 82.00 | **79.33** | 82.03 |
| KMLM$_R$-XLM-R$_{LARGE}$ (ours) | 92.81 | **76.22** | **84.12** | 78.63 | **82.95** |

The WikiAnn dataset allows us to evaluate our models on all of the 10 languages used for pretraining. As the results shown in Table 3, our base and large models improve the average performance by 2.18 and 0.32 respectively, over the corresponding XLM-R models. We notice the results on ja and zh are much lower compared with the other languages. The ja result of XLM-R$_{LARGE}$ is even worse than XLM-R$_{BASE}$. If we exclude ja and zh, the average performance improvement of our base and large models can be further increased to 3.57 and 2.01, respectively.

Table 3: Zero-shot cross-lingual NER F1 on the WikiAnn dataset.

|  | en | vi | nl | de | fr | it | es | ja | ko | zh | avg. | avg. w/o ja&zh |
|---|---|---|---|---|---|---|---|---|---|---|---|---|
| XLM-R$_{BASE}$ | 82.59 | 68.09 | 80.08 | 74.71 | 76.50 | 77.06 | 71.05 | **20.34** | 48.46 | **26.32** | 62.52 | 72.32 |
| KMLM-XLM-R$_{BASE}$ (ours) | **83.75** | **70.73** | **82.44** | **78.03** | 78.88 | **80.20** | 74.97 | 18.39 | **58.16** | 20.58 | 64.61 | **75.89** |
| KMLM$_R$-XLM-R$_{BASE}$ (ours) | 83.43 | 70.55 | 82.18 | 77.87 | **79.19** | 80.06 | **75.96** | 19.32 | 57.54 | 20.95 | **64.70** | 75.85 |
| XLM-R$_{LARGE}$ | 84.34 | 77.61 | 83.72 | 78.92 | 79.93 | 81.24 | 73.59 | **18.94** | 59.27 | **28.35** | 66.59 | 77.33 |
| KMLM$_R$-XLM-R$_{LARGE}$ (ours) | **85.07** | **77.89** | **84.55** | **81.32** | **83.65** | **82.57** | **78.93** | 14.95 | **60.68** | 19.43 | **66.91** | **79.34** |

## 4.3 MULTILINGUAL FACTUAL KNOWLEDGE RETRIEVAL

X-FACTR (Jiang et al., 2020) is a benchmark for assessing the capability of multilingual pretrained language model on capturing factual knowledge. It provides multilingual cloze-style question templates and the underlying idea is to query knowledge from the models for filling in the blank of these question templates. From (Jiang et al., 2020), we notice the performance of XLM-R$_{BASE}$ is much worse than mBERT (see Table 4). It is probably because mBERT has a much smaller vocabulary than XLM-R (120k vs 250k) and employs Wikipedia corpus instead of the general data crawled from the Internet. So we also pretrain KMLM$_R$-mBERT for more comprehensive comparison. As we can see from Table 4, all of the models trained with our framework demonstrate significant improvements on factual knowledge retrieval accuracy, which again shows that our method can help models learn more accurate factual knowledge.

Table 4: Factual knowledge retrieval results (acc., %) on X-FACTR.

|  | en | es | fr | nl | ja | ko | vi | zh | avg. |
|---|---|---|---|---|---|---|---|---|---|
| mBERT | 8.4 | 8.7 | 5.5 | 8.6 | 1.0 | 2.0 | 4.7 | 4.5 | 5.4 |
| KMLM$_R$-mBERT (ours) | **13.0** | **10.9** | **8.5** | **11.8** | **2.0** | **3.2** | **10.1** | **10.7** | **8.8** |
| XLM-R$_{BASE}$ | 4.5 | 3.1 | 2.0 | 1.6 | **1.8** | 2.1 | 3.6 | 1.0 | 2.5 |
| KMLM-XLM-R$_{BASE}$ (ours) | 8.4 | **4.8** | **4.3** | **5.7** | 1.3 | **4.5** | 5.5 | 2.5 | 4.6 |
| KMLM$_R$-XLM-R$_{BASE}$ (ours) | **8.6** | **4.8** | 4.2 | 5.6 | 1.6 | 4.2 | **5.8** | **3.0** | **4.7** |
| XLM-R$_{LARGE}$ | 7.9 | 4.4 | 3.8 | 5.0 | **2.9** | 5.2 | 5.7 | 1.0 | 4.5 |
| KMLM$_R$-XLM-R$_{LARGE}$ (ours) | **10.5** | **5.5** | **6.9** | **7.1** | 1.1 | **6.7** | 5.7 | **1.6** | **5.6** |

## 4.4 CROSS-LINGUAL RELATION CLASSIFICATION

RELX (Köksal & Özgür, 2020) is developed by selecting a subset of KBP-37 (Zhang & Wang, 2015), a commonly-used English relation classification dataset, and by generating human translations and annotations in French, German, Spanish, and Turkish. We evaluate the same set of models as §4.3, since mBERT performs better than XLM-R$_{\text{BASE}}$ also on this task. The evaluation script provided by Köksal & Özgür (2020) is used to finetune the pretrained models on English training set and evaluate on the target language test sets. As the results shown in Table 5, our base models outperform the baselines by 0.8% to 1.6% on average. The accuracy of the large model is also increased by 0.6% on average.

Table 5: Zero-shot cross-lingual relation classification performance (acc., %) on RELX.

|  | en | es | de | fr | avg. |
|---|---|---|---|---|---|
| mBERT | **65.8** | 58.9 | 58.5 | 58.2 | 60.3 |
| KMLM$_R$-mBERT (ours) | 64.2 | **59.5** | **59.1** | **61.7** | **61.1** |
| XLMR$_{\text{BASE}}$ | **62.7** | 55.1 | 54.8 | 54.3 | 56.7 |
| KMLM-XLM-R$_{\text{BASE}}$ (ours) | 60.0 | 58.0 | **57.5** | 57.8 | **58.3** |
| KMLM$_R$-XLM-R$_{\text{BASE}}$ (ours) | 60.3 | **58.7** | 54.4 | **59.4** | 58.2 |
| XLMR$_{\text{LARGE}}$ | 62.8 | 62.6 | **60.4** | 59.5 | 61.3 |
| KMLM$_R$-XLM-R$_{\text{LARGE}}$ (ours) | **63.5** | **63.7** | 60.1 | **60.4** | **61.9** |

## 4.5 CROSS-LINGUAL LOGICAL REASONING

**Dataset** We propose a cross-lingual logical reasoning (XLR) task in the form of multiple-choice questions to verify the effectiveness of our logical reasoning oriented pretraining tasks (§3.2) in an intrinsic way. An example of such reasoning question is given in Fig. 5. The dataset is constructed using the length-3 and length-4 cycles extracted from Wikidata, same as the examples shown in Fig. 2. For each cycle, we pick a triplet to create the question and answer. The question is created by asking the relation between a pair of entities in that triplet. 6 choices are provided for each question (including the correct answer), which contains all of the relations appear in the cycle and some sampled relations associated with the two entities. The remaining triplets from the cycle are used as the context, which is in the form of knowledge graph. The model is required to select the most probable choice according to the given context and question.

We manually annotate 1,050 samples in English to build the test set. The train and dev sets are automatically generated, and then cleaned by balancing the appearances of entities, relations and answers. After cleaning, we randomly select 3k train samples and 1k dev samples for the experiment. Then the multilingual test data in the other 9 non-English languages (see Sec. 4.1) are generated by selecting the entity/relation labels in the desired languages from Wikidata. The cycles used to build the test set are removed from the pretraining data, so our models have never seen them beforehand. More statistics about the dataset can be found in Appendix §A.2.

Context: (Poland, located in time zone, UTC+01:00)     (Poland, located in time zone, Central European Time)
Question: What is the relation between UTC+01:00 and Central European Time?
Answer: said to be the same as
Choices: said to be the same as, located in time zone, instance of, part of, has part, followed by

Figure 5: An example (English) extracted from our cross-lingual logic reasoning (XLR) dataset.

**Results** We modify the multiple choice evaluation script implemented by Hugging Face[4] for this experiment. The models are finetuned on the English training set, and evaluated on the test sets in different target languages. Results are presented in Table 6. All of our models outperform the baseline significantly. Moreover, KMLM$_R$-XLM-R$_{\text{BASE}}$ also outperforms KMLM-XLM-R$_{\text{BASE}}$ consistently though both of them are trained on the same data, which proves the usefulness of our proposed logical reasoning oriented pretraining task.

---

[4]https://github.com/huggingface/transformers/tree/master/examples/pytorch/multiple-choice

Table 6: Zero-shot cross-lingual logic reasoning performance (acc., %).

| | en | de | es | fr | it | ja | ko | nl | vi | zh | avg. |
|---|---|---|---|---|---|---|---|---|---|---|---|
| XLM-R$_{BASE}$ | 64.38 | 46.38 | 49.36 | 45.46 | 46.38 | 21.58 | 32.53 | 54.06 | 39.11 | 27.80 | 42.70 |
| KMLM-XLM-R$_{BASE}$ (ours) | **71.52** | 59.68 | 62.73 | 60.03 | 60.12 | 32.88 | 40.47 | 62.57 | 52.76 | 42.03 | 54.48 |
| KMLM$_R$-XLM-R$_{BASE}$ (ours) | **71.52** | **63.52** | **64.73** | **66.19** | **63.68** | **37.04** | **43.39** | **63.42** | **55.17** | **45.77** | **57.45** |
| XLM-R$_{LARGE}$ | 79.39 | 68.38 | 73.46 | 71.20 | 70.82 | 56.25 | 47.61 | 70.98 | 66.00 | 55.11 | 65.92 |
| KMLM$_R$-XLM-R$_{LARGE}$ (ours) | **87.07** | **85.87** | **83.58** | **86.03** | **83.80** | **75.04** | **75.39** | **85.01** | **83.58** | **83.74** | **82.91** |

## 4.6 General Cross-lingual Tasks

Our models are directly trained on the structured knowledge graph data. Though we attempt to minimize its difference from the natural sentences when designing the pretraining tasks, it is unknown how the difference affects cross-lingual transfer performance on the general NLP tasks. Therefore, we also evaluate our models on the part-of-speech (POS) tagging, question answering and classification tasks prepared by XTREME (Hu et al., 2020). Experimental results are shown in Table 7. Note that many of the languages covered by these tasks are not in our pretraining data, but we include all their results when computing the average performance. Overall, the performance of our models is comparable with the baselines on all of the tasks, except POS. It is because the POS task is more sensitive to the change of the training sentence structures, while the impact on the other tasks are not obvious. From these results we can say that our KMLMs achieve consistent improvements on the knowledge-intensive tasks, as shown by the experimental results in the previous subsections, without sacrificing the performance on the general NLP tasks.

Table 7: Zero-shot cross-lingual POS, QA and classification results. Note that the performance of the languages not appearing in our prepared pretraining data are also counted.

| | POS | XQuAD | MLQA | TyDi QA | XNLI | PAWS-X |
|---|---|---|---|---|---|---|
| Metrics | Acc. | F1/EM | F1/EM | F1/EM | Acc. | Acc. |
| XLM-R$_{BASE}$ | **73.71** | **69.6 / 53.9** | 64.9 / 47.2 | 45.2 / 28.5 | **73.66** | 84.80 |
| KMLM-XLM-R$_{BASE}$ (ours) | 72.69 | 69.5 / 53.4 | **65.4 / 47.3** | **49.4 / 30.9** | 73.39 | 84.47 |
| KMLM$_R$-XLM-R$_{BASE}$ (ours) | 72.76 | 69.2 / 53.3 | 64.7 / 47.0 | 48.7 / 29.2 | 73.56 | **85.07** |
| XLM-R$_{LARGE}$ | **75.25** | **76.8 / 60.9** | **72.5 / 54.2** | **66.6** / 46.6 | **78.95** | **87.81** |
| KMLM$_R$-XLM-R$_{LARGE}$ (ours) | 73.52 | 76.5 / 60.6 | 72.0 / 53.7 | 66.4 / **47.9** | 78.55 | 87.66 |

## 5 Conclusions

In this paper, we have presented a novel framework for knowledge-based multilingual language modeling. Our approach firstly creates a synthetic multilingual corpus from the existing knowledge graph and then tailor-makes two pretraining tasks, namely, multilingual knowledge oriented pretraining and logical reasoning oriented pretraining, to consolidate the knowledge modeling and multilingual pretraining and boost the capability of implicit knowledge modeling. We evaluate the proposed framework on a series of knowledge-intensive cross-lingual benchmarks and the comparison results consistently demonstrate its effectiveness.

## Reproducibility Statement

We propose a framework to pretrain a knowledge-based multilingual language model. All of the data used in our framework, including the Wikipedia/CC100 plain text and the Wikidata knowledge base, can be conveniently accessed from the Internet. Detailed hyper-parameters and training data statistics are presented in the Appendix. Except the self-constructed cross-lingual logical reasoning dataset, all of the downstream task datasets and evaluation scripts used in our experiments can be easily obtained by either following the provided URL or checking the cited papers. Our code used for model pretraining can be found in the supplementary material. We will also make the code, the pretrained language models and all data prepared by us publicly available.

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

## A  APPENDIX

### A.1  LANGUAGE MODEL PRETRAINING DETAILS

**Training Data**   The statistics of the data used for pretraining are shown in Table 8.

Table 8: Statistics of the data used for pretraining.

| Description | Value |
|---|---|
| number of languages | 10 |
| code switched synthetic sentences | 246,783,693 |
| number of unique relation combinations in length-3 cycles | 29,819 |
| number of unique relation combinations in length-4 cycles | 239,966 |
| number of reasoning based training samples from length-3 cycles | 24,142,272 |
| number of reasoning based training samples from length-4 cycles | 73,881,422 |
| number of sampled CC100 sentences (KMLM-XLM-R only) | 260,000,000 |
| number of sampled Wikipedia sentences (KMLM-mBERT only) | 153,011,930 |

**Hyper-Parameters**   The hyper-parameters used for language model pretraining are presented in Table 9. After pretraining, we finetune the models on the plain text data with max sequence length of 512 for another 600 steps.

Table 9: Hyper-parameters used for language model pretraining.

| Hyper-parameter | Value |
|---|---|
| learning rate | 5e-5 |
| weight decay | 0 |
| optimizer | AdamW |
| number of train epochs | 1 |
| batch size for the natural sentences | 9,600 |
| batch size for code switched knowledge data | 9,600 |
| batch size for reasoning data | 9,600 |
| mlm probability | 0.15 |
| max sequence length | 128 |
| number of warmup steps | 100 |
| knowledge task loss weight ($\alpha$ in the loss function) | 0.3 |

## A.2   CROSS-LINGUAL LOGIC REASONING DATA

Statistics of the self-constructed cross-lingual logic reasoning (XLR) dataset are presented in Table 10. The multilingual test data in the 9 non-English languages are generated by selecting the entity/relation labels in the desired languages from Wikidata. So the statistics for their test sets are the same as English.

Table 10: Statistics of the self-constructed cross-lingual logic reasoning data (English).

| Description | Value |
|---|---|
| number of samples in the train set | 3,000 |
| number of samples in the dev set | 1,000 |
| number of samples in the test set | 1,050 |
| train set unique relation combinations | 1,419 |
| dev set unique relation combinations | 746 |
| test set unique relation combinations | 444 |

