# OpenReview forum: "Knowledge Based Multilingual Language Model"
_ICLR.cc/2022/Conference — ICLR 2022 Submitted_

### Official Review · Reviewer_Rt8N · 2021-11-02

**Correctness:** 3
**Technical Novelty And Significance:** 2
**Empirical Novelty And Significance:** 3
**Recommendation:** 3
**Confidence:** 4

**Main Review:**

The major contribution of this work is the way of constructing synthetic sentences to encode multilingual knowledge. The code-switching is straight-forward and common in the cross-lingual and multilingual setting. And encoding logical patterns into LMs is an interesting problem and this work sampled simple “circles” from the KGs. Although sampling subgraphs is a good idea, this work’s method is still simple and does not give a clear definition of “circle”. Why are two kinds of “circles” important for logical reasoning? Could it extend to more patterns? So far, this work did not give a paradigm of formulating logical patterns from the KGs (if it is the major strength), which a bit hurts the novelty. Another weakness is that the writing is not clear. For the pre-training tasks,  what is the cross entropy loss of “L_{CS}” over the masked entity and relation tokens? And the same for the loss of “L_{L}”

Some clarification questions:
1). In all the result tables, what are the results of KMLM-XLM-R_{large}(ours)? Seems the authors did not report all the performance of KMLM-large without logical reasoning.



**Summary Of The Paper:**

     This paper proposed a new pretraining framework of incorporating multilingual knowledge into language models. The LMs are pre-trained over a large amount of synthetic code-switched sentences sampled from the multilingual KG, wikidata. Besides factual knowledge, the methods also sampled subgraphs from KG and learned logical patterns. The experiments show that the framework improved many knowledge-intensive tasks.


**Summary Of The Review:**

This work proposed to solve interesting problems of multilingual knowledge pretraining. The method is straight-forward and the pre-training task or logical patterns are not clearly written. Some ablation studies should be further investigated such as with/without normal sentences from CC100 corpus. More importantly, the author should think more about logical reasoning.

---

### Official Review · Reviewer_RyT7 · 2021-11-02

**Correctness:** 3
**Technical Novelty And Significance:** 3
**Empirical Novelty And Significance:** 3
**Recommendation:** 5
**Confidence:** 3

**Main Review:**

Strength of the paper:

The paper presents a framework to pre-train Knowledge-based Multilingual Language Models, the proposed model is evaluated on a wide range of knowledge-intensive tasks,  the knowledge-intensive training data is built on 10 languages.

The proposed model doesn’t require training from scratch, it can be initialized with checkpoints of existing pre-trained models.

Extensive experiments are performed to justify the advantage of the proposed approach;
- Cross-lingual NER: The proposed model improves F1 performance by an average of 1.75 points compared to XLM-RBASE which is an existing SOT (Table 2)
- Multilingual Factual Knowledge Retrieval: Results in Table 4 shows that the proposed method significantly improves the performance compared to other methods.
- Cross-lingual Logical Reasoning: Results in Table 6 show that the proposed model significantly outperforms the baselines.

Weakness of the paper:

The technical contributions are not explicitly highlighted, the description of the proposed approach could be made more sound, and or easy to follow with either algorithm or figure/workflow.

General Cross-lingual Tasks: The results in table 7 demonstrate that the performance of the proposed method is less compared to existing methods.


**Summary Of The Paper:**

The paper argues that existing knowledge-based language models are all trained on monolingual knowledge graph data, hence limiting their application to more languages.

To overcome this, the paper presents a framework to pre-train Knowledge-based Multilingual Language Models  (KMLMs).

The proposed framework (KMLMs);
First generates synthetic sentences (code-switched) and reasoning-based multilingual training data using Wikidata knowledge graph.
Then design pretraining tasks to facilitate knowledge learning, which allows the language models to not only memorize the factual knowledge but also learn useful logical patterns based on the intra- and inter-sentence structures of the generated data.

 KMLM is evaluated on a wide range of knowledge-intensive cross-lingual NLP tasks, including named entity recognition, factual knowledge retrieval, relation classification, and logic reasoning.

 It achieves consistent and significant improvements on all knowledge-intensive tasks, and it does not sacrifice the performance on general NLP tasks.


**Summary Of The Review:**

The paper deals with Knowledge-based Multilingual representation which is relevant and a valuable contribution to the field of language understanding.

The authors provide good and clear motivation and introduction to their research work. In general, the paper is well structured.

The discussion of related work is adequate with a number of recent citations. It could be extended in the final version, especially regarding the future directions.

The evaluation part contains interesting implementation details, used technology is state-of-the-art. The evaluation is convincing and nicely shows the merit of the approach.

Although the proposed method looks incremental, it has the following advantages:

1. The proposed model is explicitly trained to derive new knowledge through logical reasoning in addition to memorizing knowledge facts.

2. It does not require a separate encoder for knowledge graph encoding. And also doesn't rely on any entity linker to link the text to the corresponding entities, as done in existing methods.

3. It also keeps the model structure of the multilingual pre-trained language model without introducing any additional component during both training and inference stages.

Recommendation

The technical contributions could be explicitly highlighted, the description made more sound, and or easy to follow with either algorithm or figure/workflow.

Section 3.1 Reasoning Based Training Data: The authors mention that “Fig. 2 (a) the cycles of length 3 can be viewed as the basic components of complex logical reasoning process”. This is however not clear, a more clear explanation would be helpful.

English needs some proofreading.

---

> ### Comment · Reviewer_RyT7 · 2021-11-24
> **I have looked at the informative reviews from the other reviewers and I agree with most of the points raised.**
>
> I have looked at the informative reviews from the other reviewers and I agree with most of the points raised. I maintain my previous recommendation that the paper can be further improved.
> Thank you.

---

### Official Review · Reviewer_C6ht · 2021-11-04

**Correctness:** 2
**Technical Novelty And Significance:** 3
**Empirical Novelty And Significance:** 1
**Recommendation:** 3
**Confidence:** 5

**Main Review:**

While the paper presents a novel way of implicitly adding knowledge base information at pre-training time that may improve performance on knowledge related downstream tasks, it is hard to assess the impact of the approach, due to a shallow evaluation on the experiments.

Each of the experiments follows the same structure: the model is fine-tuned on English for the downstream task and zero-shot on the other languages. Therefore, the reported results for English are not zero-shot, and should be compared to the previous work that reported on the same benchmarks for supervised English. These are missing in all experiments. Moreover, there are other state-of-the-art models and approaches that could be evaluated on the same premise, by training on English and zero-shot evaluating on the other languages, including knowledge-enhanced ones cited in the paper, which would enable a fair comparison of the proposed approach with previous work.

The evaluation presented seems to point that including extra knowledge-based sentences to the pre-training phase helps in knowledge intensive downstream tasks. This is expected and the paper shows it when compared to vanilla pre-trained Transformer models. It is just hard to assess how impactful this addition is when compared to previous work that also, either implicitly or explicitly, incorporates knowledge from knowledge bases, as those described in the related work. While it is true that most of them were monolingual, they could be compared to the supervised English experiments in the paper, or evaluated using the same zero-shot setup.



**Summary Of The Paper:**

This paper presents a new language model that is enhanced with knowledge from Wikidata. The authors use two masked language modeling strategies by training with language-switched relation triples and Wikidata graph cycles (unrolled as sequences of triples). Experiments are performed in several tasks: cross-lingual NER, factual knowledge retrieval, relation classification, logical reasoning and other tasks.

**Summary Of The Review:**

An interesting idea, but with an evaluation that does not demonstrate the true impact of the proposed knowledge-enhanced model.

---

> ### Comment · Reviewer_C6ht · 2021-11-24
> **I have read several issues in the different reviews and I agree with all of the points raised.**
>
> I have read several issues in the different reviews and I agree with all of the points raised. The paper can certainly be improved in the next iteration. Thank you.

---

### Official Review · Reviewer_pMnq · 2021-11-05

**Correctness:** 4
**Technical Novelty And Significance:** 2
**Empirical Novelty And Significance:** 3
**Recommendation:** 5
**Confidence:** 3

**Main Review:**

Strengths:
1. The paper is well-written and well-presented.
2. The experiments are extensive and convincing.
3. The model shows strong performance on a wide variety of tasks that is knowledge-intensive.

Weakness:
1. The experiments done are mainly in a zero-shot setting, while impressive, there might be concerns regarding finetuning performance compared with existing baselines.
2. The idea of code-switching in multilingual knowledge is not completely new, for example, X-Factr is also doing code-switching in the prompts to better retrieve factual knowledge. This should be further clarified in the paper contribution.
3. I think the sampling of synthetic data should be discussed more, since this is the focus of the paper pretraining method. More ablation on how to sample and handle the problems of distribution mismatch between true texts vs. template texts could shed light on what setting such pretraining would be most useful.
4. Some focus should be given to more diverse sets of languages, esp. low-resource languages. Since it's a multi-lingual model and utilizes code-switching, I wonder if it can increase performance for some low-resource language tasks by leveraging connections to popular languages.

**Summary Of The Paper:**

The paper presents a new pre-trained language model that works on multiple languages. The language model pretraining has a special focus on incorporating knowledge, by generating a large amount of code-switched synthetic sentences and reasoning-based multilingual training data using the Wikidata knowledge graphs. The language model not only memorizes the factual knowledge but also learns useful logical patterns. Experiments on a wide range of knowledge-intensive cross-lingual tasks show performance improvemnents. The author also proposes a new task to test the model's ability of logical reasoning.

**Summary Of The Review:**

The paper is having really good performance across multiple tasks and is simple to implement and follow. The multilingual is a nice addition to existing knowledge-based LMs, but there are some concerns regarding the pretraining data generation and low-resource setting.

---

> ### Comment · Reviewer_pMnq · 2021-11-24
> **I have read other's review and I maintain my Recommendation score**
>
> I have read others' reviews and I agree with most points other reviews have pointed out and I maintain my Recommendation score. Thank you.

---

### Official Review · Reviewer_cZgu · 2021-11-05

**Correctness:** 3
**Technical Novelty And Significance:** 2
**Empirical Novelty And Significance:** 2
**Recommendation:** 5
**Confidence:** 4

**Main Review:**

### Strengths of the paper:

- This addresses an important area in large language models, of better encoding knowledge in multilingual models. I believe this is a useful direction and will be of interest to the NLP community at ICLR.
- I think the idea of generating code-switched synthetic sentences from KG triples to be really nice. If it could be shown that this technique can be used to boost performance of low-resource languages by leveraging KG resources from English, I believe this technique could be of practical use. It may also motivate us to contribute aliases to Wikidata for low resource languages.
- The idea of the edge/relation completion task is also quite interesting. This task seems related to a few-shot/multiple choice relation classification task  but where the task presents 2 related pieces of information (KG triples), instead of a single (natural language) triple.

### Weaknesses of the paper:

- My main criticisms of this work stems from a single point, which is, after reading the paper, it was not exactly clear what the main problem being addressed. Although the techniques and experiments presented were thematically related, in that they used synthetic sentences derived from WikiData, it was not quite clear that they were addressing a specific problem. For example: the code-switched data augmentation idea seems to be related to the question “can we better perform knowledge transfer between languages using multilingual KG” or perhaps “using multilingual KG can augment knowledge in LMs since certain knowledge is not covered in Wikipedia/Web documents.”, or some other specific hypothesis?  Given a better defined hypothesis, it may be easier to evaluate how much of the problem is solved by the techniques in this paper.

  The “logical reasoning” task, OTOH, seems to be somewhat independent of cross- or multi- linguality? Is the hypothesis that the logic reasoning task allows LMs to better perform edge/relation prediction? Is there anything specifically cross-lingual about learning this task, or would one be able to learn/evaluate its effectiveness in a mono-lingual setting?


- Related to the above: the cross-lingual experiments show relatively modest performance wins across the different tasks evaluated. It raises the question of whether there is something specific about code-switching the triples that are helpful or whether the extra KG data, even in mono-lingual format, would have been sufficient to improve performance. Given that previous works have shown that multilingual LMs are able to transfer tasks across languages,  I think it would have been very useful to train a baseline that contains all the KG triples used in the KLML-variants, but simply with monolingual versions of the triples.

  As an example of this, see the results from X-FACTR for the English language which saw the largest improvement in performance. It seems that the improvement may have come purely from allowing the model to train on KG triples, and is potentially unrelated any cross-lingual transfer.

### Other comments:

- While I thought the “cross-lingual logical reasoning” pretraining task to be an interesting one, I do believe the evaluation needs a bit more work. The current evaluation, as presented in Section 4.5, is quite close to the pretraining task in both content and format. As such, it feels as this result is not as “zero-shot” and it's a bit hard to conclude whether gains are obtained from some cross-lingual transfer or whether its from a large overlap in pretraining and evaluation tasks.

- Also related to the logical reasoning task: besides the intrinsic evaluation in Section 4.5, none of the other evaluated tasks saw much improvements. Is there an expectation that this pretraining objective could be really helpful on some other task? Which ones?
In a few of the results tables, particularly ones marked as “zero-shot” it is unclear what exactly the training setup is. For example Tables 2 and 3 claim to be zero-shot NER, however the text describes supervised training with Conll 02/03 in English. Is English in Table 2 still zero-shot or supervised? What about Table 3 (WikiAnn)? Is the model trained on CoNLL and evaluated on WikiAnn? Or is any WikiAnn training data used?

- For X-FACTR it's not clear what is the training/fine-tuning setup. It doesn’t say which languages are used for training.

- For RELX, we see the same problem as NER: it seems that the model is trained on English and evaluated on other languages? Please make this clear.

- For XLR, please also make a note on the table that English is *not* zero-shot, but fully supervised.

- Also for the XLR it was unclear how exactly the multiple choice task was formulated. While there is a reference to a HuggingFace script, it is a bit obscure (and the script can be removed at any time). It would be better to simply describe how the multiple choice task is modeled here.

- Perhaps a more direct evaluation of improved knowledge representation in the KLML model is the “Multilingual LAMA” task from :
    1.  “Multilingual LAMA: Investigating Knowledge in Multilingual Pretrained Language Models” https://aclanthology.org/2021.eacl-main.284.pdf , https://github.com/norakassner/mlama

- Also, related to training on KG triples, I think it would have been useful to compare with the work of verbalizing KG triples as in:
  1. “The WebNLG Challenge: Generating Text from RDF Data” https://aclanthology.org/W17-3518/  , https://webnlg-challenge.loria.fr/challenge_2020/
  2. “Knowledge Graph Based Synthetic Corpus Generation for Knowledge-Enhanced Language Model Pre-training”, https://aclanthology.org/2021.naacl-main.278.pdf , https://github.com/google-research-datasets/KELM-corpus


**Summary Of The Paper:**

This paper introduces two new techniques for pretraining large multilingual language models focusing on knowledge intensive tasks. It first proposes generating code-switched synthetic sentences derived from triples from a knowledge graph (WikiData) and subsequently using these synthetic sentences as input to a multilingual masked language model. In addition, the work proposes using cycles in knowledge graph nodes, it proposes a new task that uses the derived synthetic sentences and asks the model to predict (and learn from) a masked-out relation (edge). The paper demonstrates the efficacy of the new pretraining methods using a new cross- and mult- lingual tasks, including: NER (Conll 02/03), fact retrieval (X-FACTR), relation extraction (RELX), and some XTREME tasks. In addition, they describe a new cross-lingual logical reasoning task (XLR) to verify the effectiveness of the “logical reasoning” pretraining task.


**Summary Of The Review:**

This addresses an important area in large language models, of better encoding knowledge in multilingual models. I believe this is a useful direction and will be of interest to the NLP community at ICLR. I think the two core techniques proposed here (code-switching using KG triples and using KG cycles to teach models about transitive relations) are new and interesting. The main drawback of this work is that the improvements on the tasks chosen are quite modest. It is also unclear whether a simpler version of the proposed technique (discussed above, of training using monolingual triples) would have been sufficient for these modest gains. I think the work would benefit from more clearly defining the main hypotheses in the study and more carefully evaluating (with also qualitative analysis) the benefits of the proposed techniques.

---

### Comment · Area_Chair_t2KS · 2021-11-14
**Additional Discussion Encouraged**

Dear Reviewers,

can you please take a look at each other's reviews? Your reviews currently straddle the decision boundary and it would be good to make sure you have considered all the perspectives provided. Please update your reviews (at least to acknowledge that you have read all reviews).

Thanks,
Your Area Chair

---

### Decision · Program_Chairs · 2022-01-20

**Decision:**

Reject

**Comment:**

This paper studies the important problem of adding structured knowledge (in this case from Wikidata) to pretrained language models. The reviewers do not see this paper as ready for ICLR and recommend a number of revisions. Unfortunately the authors did not respond during the author response period. The area chair hence agrees with the reviewers.